# REAL: Benchmarking Autonomous Agents on Deterministic Simulations of Real Websites

Divyansh Garg[1][*]   Diego Caples[1]   Andis Draguns[5,6]   Nikil Ravi[2]   Pranav Putta[7]
Naman Garg[1]   Prannay Hebbar[1]   Youngchul Joo[1]   Jindong Gu[3]   Charles London[3]
Christian Schroeder de Witt[3]   Sumeet Motwani[3]

[1]The AGI Company   [2]Stanford University   [3]University of Oxford
[5]Contramont Research   [6]IMCS UL   [7]Plato

## Abstract

We introduce **REAL**, a benchmark and framework for multi-turn agent evaluations on deterministic simulations of real-world websites. REAL comprises high-fidelity, publicly hosted, deterministic replicas of 11 widely-used websites across domains such as e-commerce, travel, communication, and professional networking. We also release a benchmark consisting of 112 practical tasks that mirror everyday complex user interactions requiring both accurate information retrieval and state-changing actions. All interactions occur within this fully controlled setting, eliminating safety risks and enabling robust, reproducible evaluation of agent capability and reliability. REAL environments are highly configurable, offer complete action/observation space control, and allow researchers to inspect state-changes at any step to define reward signals for training. Our novel evaluation framework combines programmatic checks of website state for action-based tasks with rubric-guided LLM-based judgments for information retrieval, and our harness supports both open-source and proprietary agentic systems. Our empirical results show that frontier language models achieve at most a $41\%$ success rate on REAL, highlighting critical gaps in current autonomous capabilities. REAL enables easy integration of new tasks, reproducible evaluation, and scalable data generation for post-training web agents. The websites, framework, and leaderboard are available at `https://realevals.xyz` and `https://github.com/agi-inc/REAL`.

## 1 Introduction

Large Language Models have demonstrated remarkable advances in reasoning capabilities, suggesting a path toward human-level performance across domains [20, 4]. Agents leveraging these models promise to automate countless routine digital tasks with substantial economic impact [7], yet consistently struggle with reliably executing multi-turn web interactions that most humans complete effortlessly [50]. Real-world deployment has been slow despite general capability improvements, and can be attributed to the lack of adequate real-world web based training and evaluation environments. This gap impedes research progress and delays the usefulness of reliably functioning web-agents.

Current methods for evaluating web agents face several fundamental limitations. First, real websites lack determinism, with constantly changing underlying data along with evolving UX workflows, making reproducible evaluation nearly impossible. Second, production websites cannot be configured to test critical edge cases, such as out-of-stock items, network latency variations, or error recovery scenarios [9]. Third, agents may change the state of the website themselves (via payments and state-changes), raising concerns of safety, costs, and robustness during evaluation. Prior works [52, 60] have made valuable progress but introduce artificial constraints such as heavily restricted

---

[*]Co-first author with Shaun VanWeelden; correct author list reflected in the public arXiv version.

39th Conference on Neural Information Processing Systems (NeurIPS 2025) Track on Datasets and Benchmarks..

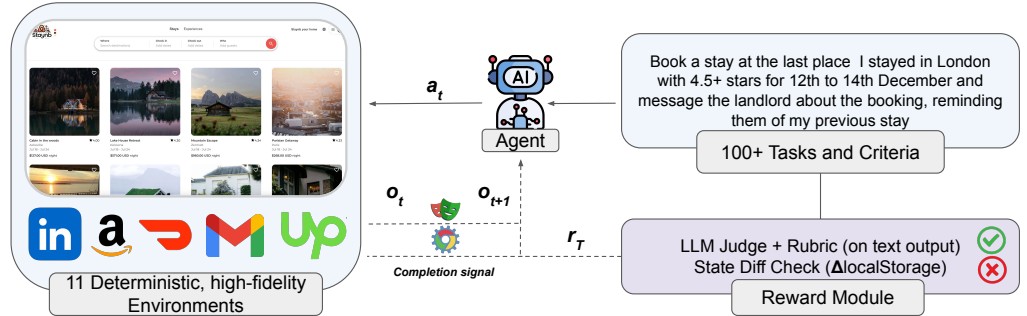

Figure 1: **The REAL benchmark and framework.** REAL provides 11 realistic, deterministic, high-fidelity web environments (across e-commerce, networking, communication, scheduling, booking, project management) and 110+ evaluation tasks. An agent interacting with the environments receives an observation ($o_t$) and executes actions ($a_t$) to complete a task. Upon completion, an outcome reward ($r_T$) is evaluated via programmatic state verification and/or a rubric based LLM-judge.

action/observation spaces, simplified tasks and interfaces that may not reflect real-world website complexity [55], and a lack of mechanisms to systematically test edge-cases. Moreover, these benchmarks are challenging to use as training environments due to the difficulty of defining clear reward signals or observing state-diffs after actions. Such limitations have created a *systemic gap between benchmarks and the true challenges of autonomous reliable web navigation.*

To address these limitations, we present **REAL**, a benchmark and evaluation framework designed to test web agents on high-fidelity, deterministic replicas of popular websites. Our approach makes several key advances. First, inspired by WebArena [60], we develop accurate representations of 11 widely-used websites (across e-commerce, travel, social media, scheduling) using modern web-dev standards. These websites span several pages and mimic the visual and functional fidelity of important real-world websites. We host the sites, reducing the cost and difficulty of self-hosting benchmarks. Our environments are made deterministic by fixing all data, timestamps, and UX elements, while retaining configurability via URL parameters. This enables reproducible testing of various edge cases (latency, errors, behaviors), with website state stored in browser local storage for persistence.

REAL provides a flexible test harness for open-source and proprietary agents, offering unrestricted browser state access without fixed action/observation spaces. This design reflects the current research landscape, where approaches ranging from open APIs [45] to proprietary black-box systems [10] work with custom observation and action spaces. In line with this, we do not impose any explicit restrictions, allowing agents to communicate with the browser via Playwright for simplicity or Chrome DevTools Protocol (CDP)[2] for complete control over the session.

For evaluations, we provide practical information retrieval and state-changing tasks, initiated by a natural language user request along with the website configuration URL. The REAL framework allocates a persistent CDP session to the agent, enabling low level browser automation while maintaining state throughout the interaction. When an agent marks the task as complete, it triggers the capture of the local storage changes and the model's output. Performance is evaluated via two methods: (1) programmatic comparison of pre-task and post-task local storage states for action-oriented tasks; and (2) a structured LLM-judge using task-specific rubrics for information retrieval tasks [62]. We evaluate frontier models with a baseline agent that we provide as part of REAL. Our current evaluations indicate that no model achieves more than $41.07\%$ performance on our tasks, with Claude 3.7-Sonnet, Gemini 2.5 Pro, and GPT-4o achieving $41.07\%$, $38.39\%$, and $14.29\%$ respectively (see Section 7).

In this paper, we provide a detailed description of the current state of agentic systems and benchmarks (Sec. 2), our web environments (Sec. 3), how agents can use these environments (Sec. 4), our task design and evaluation methodology (Sec. 5), baseline experimental results (Sec. 7), and implications for future research (Sec. 8). Our key contributions include: (1) a collection of 11 deterministic, configurable, high-fidelity simulated web-environments; (2) a flexible evaluation framework supporting both open and proprietary agent systems; (3) a comprehensive set of 112 real-world challenges; (4) a robust evaluation method for each task along with reward signals that

---

[2]`https://playwright.dev/` and `https://chromedevtools.github.io/devtools-protocol/`

could be used for training or synthetic trajectory generation; and (5) an open leaderboard with hosted environments, making agentic evaluations easily accessible. REAL represents a significant step toward guiding the development and evaluation of highly-capable and reliable real-world web-agents.

## 2   Motivation and Related Work

**Benchmarks for Web Agents.**   Recent advances in large language models (LLMs) have led to growing interest in web agent benchmarks that evaluate an agent's ability to interact with browser-based environments. Early efforts like MiniWoB [42] and MiniWoB++ [31] introduced controlled settings and metrics for such evaluations. WebShop [52] extended this to e-commerce tasks in a single-store setting, while Mind2Web [13] released a broader dataset of over open-ended tasks. These benchmarks allow for reproducible evaluation on predefined, offline datasets.

Various works have also proposed suites of simulated web environments, e.g. WebArena [60] and VisualWebArena [23]. WebArena struggles with realism and task utility, where certain tasks involve artificially constrained ambiguous goals or actions that do not reflect everyday web usage [21]. Moreover, the benchmark requires dedicated hosting infrastructure and overhead, and the environments can be "gamed" [44] by exploiting shortcuts unavailable in real scenarios. BrowserGym [9] offers a unified interface for evaluating agents across multiple existing benchmarks through a standardized observation and action space. BrowserGym forms the foundation for our REAL implementation, which extends its capabilities to address gaps in prior benchmarks (simplified HTML structures, lack of configurability/reproducibility, tasks that do not fully reflect real-world use cases [21]).

Beyond these specialized benchmarks, efforts to create real-time or live evaluation settings face reproducibility challenges [35]. Live websites may change over time, break existing agent behaviors, or introduce unpredictable failures. In contrast, synthetic environments offer stability but often lack the realism and complexity of actual websites, leading to overfitting and poor generalization [29]. This gap underscores the unmet need for deterministic, high-fidelity, and readily accessible web benchmarks that support multiple configurations, capture genuine real-world scenarios, and act as robust testbeds for RL-based agent research, similar to OpenAI Gym [6].

**Web Agents and Post-training.**   A large portion of modern work and everyday tasks is conducted via web-based tools: filling forms, updating dashboards, making resevrations, ordering items, or navigating internal portals. Automating even a fraction of these workflows would result in massive economic productivity [7, 4]. a number of promising agents have been developed by leveraging LLM reasoning and planning capabilities in these domains [53, 57]. AgentQ [40] leverages guided MCTS combined with self-critique and iterative post-training to boost multi-step reasoning in complex web navigation tasks. OpenAI's Operator [39] and Anthropic's Computer-Use [2] employ the companies' respective models to be able to execute simple browser tasks by simulating mouse and keyboard inputs. AgentOccam [51] improves web task performance by aligning its action and observation spaces with those in the pre-training data. Several other works along these lines attempt to use exploration and planning to boost performance  [58, 24, 16].

Despite these advances, current agents are still restricted to narrow tasks and have limited error recovery mechanisms, relying on brittle prompts and struggling with complex workflows [55]. A key reason is the lack of robust training and evaluation environments. REAL addresses this by introducing the first benchmark to combine realistic, high-fidelity web simulations with configurability and deterministim, enabling safe, reproducible testing under real-world conditions. REAL's configurability and determinism make it well-suited not only for evaluation but also for reinforcement learning (RL)-based training. Recent models like OpenAI's o1 [38] and DeepSeek-R1 [12] have demonstrated the value of RL and test-time compute in boosting reasoning abilities. As RL becomes central to the advancement of agentic performance, the need for suitable training environments is also growing rapidly. REAL supports this paradigm by providing deterministic environments, state tracking, and reward signals, making it suitable for both evaluation and for RL-based web agent training.

## 3   REAL Websites

REAL consists of 11 high-fidelity realistic website implementations that replicate the workflows, functionality, and interfaces of important and widely-used consumer platforms. We highlight the

selection and development process along with several important advantages of our website environments below. All environments are created under fair use principles for scientific advancement; see Appendix A.1 for our complete legal disclaimer and fair use statement.

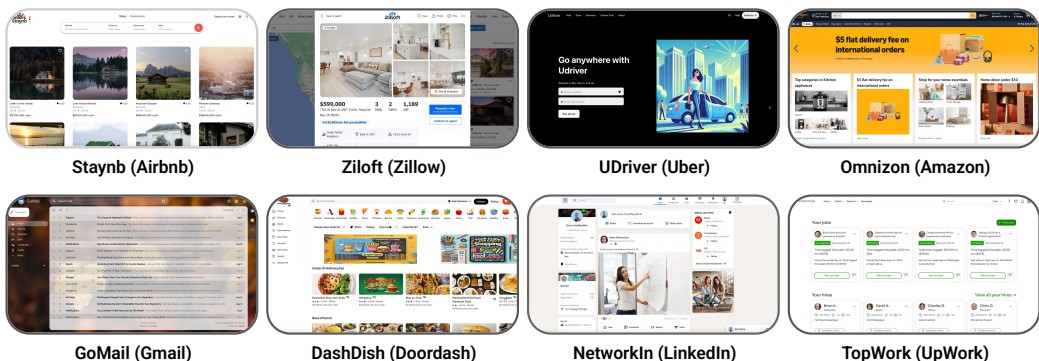

Figure 2: Screenshots of representative web environments included in REAL (8 of 11 shown). These are high-fidelity, deterministic simulations of popular websites, hosted by us for easy accessibility. These environments feature complex, multi-page workflows with persistent state management on the browser, allowing detailed tracking and inspection of state changes induced by agent actions.

**Website Selection and Development Stack.** Our website selection process focuses on a diverse set of consumer-facing applications that drive significant web traffic and economic activity [54, 11, 17]. We prioritize websites requiring varied interaction capabilities: form completion, reliable online payments, multi-step workflows, dropdown menus, map interfaces, data filtering, information retrieval, and state-dependent elements. The final collection, presented and linked in Table 1, spans key domains including e-commerce, travel, communication, scheduling, freelance marketplaces, property search, etc. This approach ensures agents are evaluated on a representative range of everyday web interactions—from airline seat selection to event scheduling and payment management—allowing for a comprehensive systematic assessment of their performance on crucial real-world tasks. Our websites are implemented using a modern front-end stack centered on React and Next.js, and all benchmark environments are publicly hosted, reducing the barrier to entry. We encourage readers to test our public environments linked in Table 1. See Appendix A.3 for more details on our tech stack.

**Determinism.** To ensure reproducible evaluations, in line with [60, 9], our websites are designed to be fully deterministic (yet highliy configurable, see Sec. 6) through several key features:

1. **Static Data**: All potentially variable information, such as product prices, availability statuses, and displayed messages, is fixed. This eliminates variability between task executions. AI-generated synthetic data was utilized where appropriate to maintain realism.

2. **Predefined Temporal Settings**: Time-dependent elements, including date selectors and time zones, are locked to guarantee consistency across all task runs.

3. **Replayability**: As a result, identical task conditions can be reliably recreated, facilitating systematic performance comparisons across different agents and experimental configs.

**Website Authentication and Browser State Management.** To streamline agent interaction, websites are pre-authenticated, bypassing standard logins for immediate access to task-specific functionalities. Common anti-automation mechanisms like CAPTCHAs are intentionally removed, similar to [60]. Website state persists across interactions (including page navigation, refreshes, and multi-tab usage) via the browser's localStorage. This ensures data continuity, mirrors realistic user session behavior, and enables agents to manage stateful tasks [40]. Local sessions can be conveniently cleared by navigating the URL to '/clear'.

| NAME | INSPIRED BY | LIVE REAL URL | CORE FUNCTIONALITY |
|---|---|---|---|
| Staynb | Airbnb | evals-staynb | Search, filter, book, and review vacation rentals; manage bookings. |
| Omnizon | Amazon | evals-omnizon | Browse/search products, manage shopping cart, complete online purchase checkout. |
| DashDish | Doordash | evals-dashdish | Browse restaurants, customize menu selections, place and manage food delivery orders. |
| GoCalendar | GCal | evals-gocalendar | Manage calendar views, schedule events, create and modify appointments. |
| GoMail | Gmail | evals-gomail | Manage inbox (read, label, delete), compose/send emails, handle attachments. |
| OpenDining | OpenTable | evals-opendining | Search restaurant availability by criteria (time, party size), make/manage table reservations. |
| NetworkIn | LinkedIn | evals-networkin | Manage user profile, search for professional connections, view profiles and posts. |
| UDriver | Uber | evals-udriver | Plan trips (set locations), request rides based on service type, view route and fare estimates. |
| FlyUnified | United | evals-fly-unified | Search for flights (origin, destination, dates), select seats, book tickets, manage itineraries. |
| TopWork | UpWork | evals-topwork | Post jobs (client), search/apply for projects (freelancer), manage proposals and active contracts. |
| Zilloft | Zillow | evals-zilloft | Search/filter property listings, save favorites, contact managers, view property details and photos. |

Table 1: **REAL Website Replicas**: High-fidelity, deterministic envs of popular sites built with modern web frameworks (React, Next.js) for reproducible evaluation of autonomous web agents.

## 4 REAL Framework and Environments

We model agent interaction within REAL environments as a Partially Observable Markov Decision Process (POMDP). The underlying environment state $s_t \in S$ encompasses the complete browser state at timestep $t$. State transitions $T : S \times A \to S$ are deterministic, governed by the browser engine executing the website's code in response to agent actions $a_t \in A$. REAL allows agents to interact in two primary ways: high-level interactions using Playwright or lower-level control via the Chrome DevTools protocol. The observation and action space for both of these modes is defined in the following section. At each step $t$, the agent receives observation $o_t \in O$, selects action $a_t$ conditioned on the task $i$ and potentially the history $(o_1^t, a_1^{t-1})$, leading deterministically to the next state $s_{t+1}$ and observation $o_{t+1}$. Task success or failure is determined by an outcome reward function $r$, evaluated at the final timestep $T$. In Section B, we describe the agent harness and evaluation flow.

**Observation Space.** REAL offers configurable observation spaces $O(s_t) \to o_t$ which can be specified based on an agent's chosen interaction modality (high-level Playwright or low-level CDP).

For agents interacting via the high-level Playwright interface, we provide default agents that can be configured to use an observation space $O$ including one or more of the following components: *Screenshots*, visual renderings of the current web page; *Full DOM*, the complete Document Object Model structure of the page; *Accessibility Tree*, a representation of the page structure based on accessibility APIs, providing semantic information about elements. This is broadly consistent with other web benchmarks [10, 40, 24, 51]. Alternatively, REAL provides the agent with direct access to the Playwright Browser object itself, allowing the use of information derivable through the Playwright API as its observation space.[3]

For agents requiring fine-grained control through the low-level Chrome DevTools Protocol (CDP), the observation space encompasses the entire live browser session state accessible via the CDP connection. This provides maximum flexibility, allowing the agent to observe any aspect available as

---

[3]`https://playwright.dev/docs/api/class-browser`

part of the browser's current session.[4] This flexibility enables researchers to adapt the observation space to the specific input requirements and capabilities of custom agent architectures or scaffolds.

**Tasks and Action Space.**   The definition of $A$ is flexible and dependent on the interaction mode.

When using the Playwright interface, the action space $A$ consists of high-level commands simulating standard user inputs. This includes but is not limited to operations such as text input, mouse clicks, keyboard commands and shortcuts, file uploads, focus elements, drag and drop, and scrolling.[5] This allows agents designed around user-level actions to operate naturally within REAL environments.

Agents interfacing with environments via CDP have access to a substantially broader action space. This low-level control permits a wide range of interactions directly within the browser environment. For instance, agents can execute commands for direct Document Object Model (DOM) modification, arbitrary JavaScript execution within the page's context, performance profiling, emulation of different devices or network conditions, interception and modification of network requests, and even detailed browser session debugging using tools like breakpoints.

**Rewards.**   Our framework primarily uses an outcome reward function $r \in \{0, 1\}$ to evaluate task success upon completion (at timestep $T$). This binary outcome reward indicates whether the agent successfully achieved the specified task goal $i$ and is determined as follows:

- **Action-based Tasks ($r_A$):** Rewards are determined by programmatic verification function $f_{eval}$, which compares the difference between the initial ($s_0$) and final ($s_T$) 'localStorage' states against a set of predefined key-value assertions specific to the task goal $i$. $r_A = 1$ if and only if all assertions pass with an exact match.

- **Information Retrieval Tasks ($r_R$):** Rewards are determined by an LLM-judge evaluation function $g_{eval}$, which assesses the agent's final submitted text response against a pre-determined task-specific rubric. $r_R = 1$ if the response is judged as correct.

- **Combined Tasks:** Require both $r_A = 1$ and $r_R = 1$ for the overall task reward $r$ to be 1.

We note that while the current version of REAL provides binary outcome rewards, the underlying framework components (deterministic environment, state tracking via 'localStorage', programmatic checks) are flexible enough to support the definition and use of dense, step-wise reward functions for reinforcement learning [30, 40]. REAL already supports partial-credit scoring via subtask checkpoints. More than half of our tasks (63/112) have a multi-subtask structure where each task's JSON defines the subtask rewards and by default logs granular subtask completion rewards.

**Evaluation Functions.**   REAL offers endpoints for evaluations, debugging, and configuration:

- `/config`: Used to initialize the environment for a specific task run. Appending query parameters to this endpoint allows setting both universal and website-specific configurations (detailed in Section 6), such as simulated `latency`, error mode flags (e.g., `error_finding_driver`), accessibility settings (`hide_aria_labels`), and run identifiers (`run_id`, `task_id`).

- `/submit`: The agent must navigate to `/submit` to signal task completion for leaderboard submissions. This action captures the final `localStorage` state and the agent's textual response. This captured data is then used by the evaluation harness to compute the reward $r$ and record the result on the public leaderboard associated with the provided `run_id`.

- `/finish`: Whenever the website state changes, those changes are saved in the website `localStorage` state. Navigating to `/finish` at any point displays the difference between the initial state and current state, allowing users to inspect the precise state changes.

- `/clear`: Navigating to `/clear` resets the website's `localStorage` to its default state.

---

[4] `https://chromedevtools.github.io/devtools-protocol/`
[5] `https://playwright.dev/docs/input`

# 5    Evaluation Tasks

REAL consists of a suite of 112 evaluation tasks across 11 website environments, designed to assess agent performance on realistic, multi-turn interactions that mirror common user goals and workflows. These tasks extend beyond simple atomic actions and are assigned difficulty levels (easy, medium, hard), reflecting factors like required planning, interaction steps, constraints, or reasoning depth. Involving both information seeking and state manipulation (approaches also employed by [52, 60, 23, 56, 32]), each task is initiated by a natural language instruction (the 'goal') for the agent. This goal may be accompanied by specific environment configurations set via the '/config' endpoint (described in Section 6). We categorize tasks as follows.

**Information Retrieval Tasks.**    Information Retrieval tasks require the agent to navigate an environment, locate specific pieces of information, potentially merge findings from multiple locations, and report the result [13, 60]. Goals range from simple single-page lookups (e.g., identifying initial listed items, finding a flight time) to complex queries demanding cross-page navigation or constraint-based filtering (e.g., finding the number of restaurants matching a specific category, summarizing event counts across different calendars for a given month). For these tasks, evaluation is based on the agent's final text response (submitted via the '/submit' endpoint). An LLM judge [59] then assesses this response against a predefined, task-specific 'rubric' for accuracy and completeness relative to the environmental ground truth.

**Action-based Tasks.**    Action-based tasks require agents to modify the environment's state, representing common goal-oriented web usage such as booking flights, scheduling calendar events, professional networking, etc. These tasks often require interpreting complex instructions involving multiple constraints (e.g., specific dates, times, locations, passenger numbers, item types, payment details) [52]. Evaluation of these tasks relies on programmatic verification of the final website state, captured via browser 'localStorage' when the agent navigates to '/submit'. We use 'state-check' mechanisms to inspect the difference between the initial and final 'localStorage' state. A task is considered complete only if all specified state conditions are met. This provides an objective and deterministic measure of the agent's ability to effect precise state changes.

**Combined Tasks and Additional Details.**    Several REAL tasks combine retrieval and action elements (identified as 'challengeType: retrieval-action'); for instance, finding an item's price, adding it to cart, purchasing, then reporting the final cost. Similar to [60], we also include tasks designed to be impossible under the given deterministic conditions, such as attempting to book non-existent flights or using invalid payment information. These tasks assess an agent's ability to recognize failure conditions, potentially employ error recovery (if applicable), and accurately report non-completion, rather than hallucinating success or failing silently [29, 41, 22]. Collectively, our evaluations test important aspects of agent performance and reliability in simulated real-world scenarios.

**Agent Harness.**    To allow straightforward integration and evaluation of various agent implementations, we provide the REAL Agent Harness, a flexible interface that reduces the implementation overhead for researchers. The harness supports three integration methods: high-level Playwright API access for standard web interactions, low-level CDP for fine-grained control, and URL-based endpoints for black-box systems. This accommodates various agent designs from simple action-based models to reasoning models without substantial modifications to existing code. When evaluating tasks, our harness manages browser instances, state tracking, and processes task completion signals. It captures critical state changes and text responses for evaluations, providing consistent metrics across different agent architectures. We provide a detailed specification of our agent harness in Appendix B.

# 6    Configurable Environments

REAL incorporates a configuration framework that enables precise control over testing conditions, significantly improving its utility for rigorous agent evaluations. Addressing limitations of static environments found in prior benchmarks [52, 60, 33] and the non-reproducibility of live websites [21], REAL implements a two-level configuration system—universal and website-specific. This structure supports systematic evaluations to develop reliable agents [9, 14], while maintaining the

determinism crucial for reproducible results. Configurations are applied for each task run via standard query string parameters appended to a dedicated /config URL endpoint on each website.

Universal configurations apply globally across all websites within the benchmark, and are used to establish consistent baseline conditions. Parameters at this level include settings such as simulated network `latency`, the `hide_aria_labels` flag to control the presence of ARIA attributes for accessibility testing, and identifiers for experiment management (`run_id`, `task_id`).

Website-specific configurations allow control over the internal state, behavior, and backend processes tailored to individual environments. This is essential for simulating specific operational scenarios, user contexts, and edge cases. Beyond initializing basic states, for e.g. the `total_conversations` on GoMail, these parameters provide detailed control relevant to real-world usage [26, 50]. We use the UDriver environment as an example of the site-specific parameters researchers can configure:

- **Introduce controlled error states** to evaluate agent error detection and recovery capabilities (e.g., setting `error_finding_driver=true` or `error_booking_ride=true`).
- **Modify the latency of operations** to assess agents under different response times (e.g., adjusting `simulating_searching_driver_delay=true`.
- **Modify application-specific logic parameters**, such as internal pricing calculations or discount availability (e.g., modifying `udriverx_multiplier` or `comfort_discount`).
- **Set initial content or regional contexts** via data presets (e.g., using `location_preset=2` to initialize the environment with data relevant to New York).

This dual-level configuration system in REAL provides researchers an extensive amount of control over specific experimental variables within a deterministic framework. Detailed configurations for each environment part of REAL are provided on our website[6].

## 7 Leaderboard

We evaluate our baseline agent with a large set of frontier models on all REAL environments. This section presents the quantitative performance and discusses some important observations derived from analyzing agent interaction trajectories.

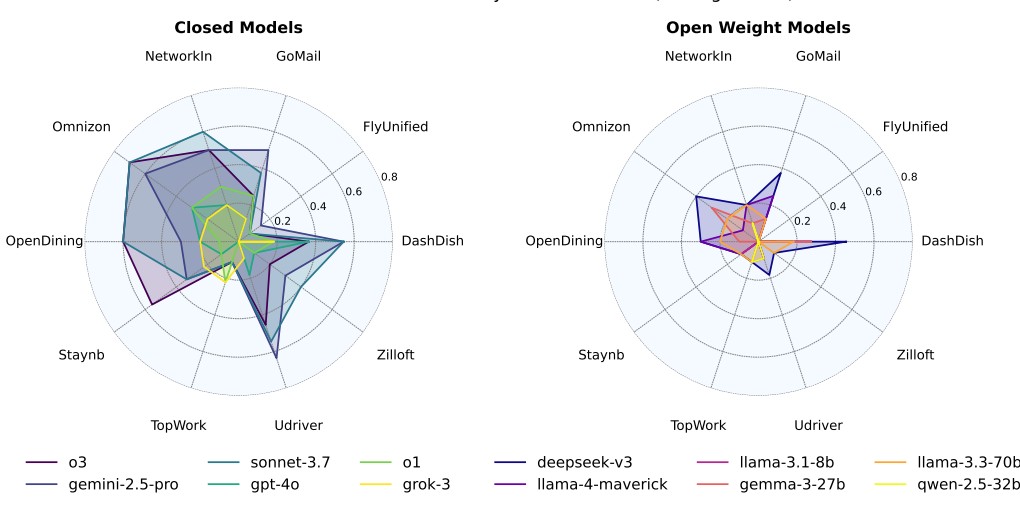

Figure 3: A per-website performance breakdown for several frontier models across REAL environments. TopWork and FlyUnified are consistently the most challenging environments.

The overall end-to-end task success rates across the 112 REAL tasks for various models are summarized in Figures 3 and 4. Performance varied considerably across models. The current leading

---

[6]See `https://www.realevals.xyz/websites/udriver` with the appropriate site name.

model is Claude-3.7-Sonnet-Thinking, achieving a success rate of 41.07%, followed by Gemini-2.5-Pro-Experimental at 38.39%. Similarly, other reasoning models also perform much better than standard pre-trained models, for example o3 (34.82%), o3-mini (25.00%), and o1 (16.07%). Despite this, there is a significant room for improvement. Open source models currently lag behind, with Llama-4-Maverick (12.50%) showing effectively similar performance to Llama 3.3 70B (10.71%) [15]. This suggests that increases in scale alone, at least between these specific models, did not translate into improved practical web navigation capabilities. Notably, DeepSeek V3 (19.64%) [12] show much better performance than Llama models. Small models lag significantly, with Llama-3.1-8B, Qwen-2.5-vl-32B, and Gemma-3-27B achieving only 1.79%, 2.68%, and 9.82% respectively, underscoring the need for substantial model capacity and training to handle the complexities of agentic performance. We also evaluated agent scaffolds such as Anthropic's Computer Use Agent, which recorded 42.90% and OpenAI's CUA model [10], which recorded 7.14% (See Sec. B.5 for more agent scaffold evaluation results).

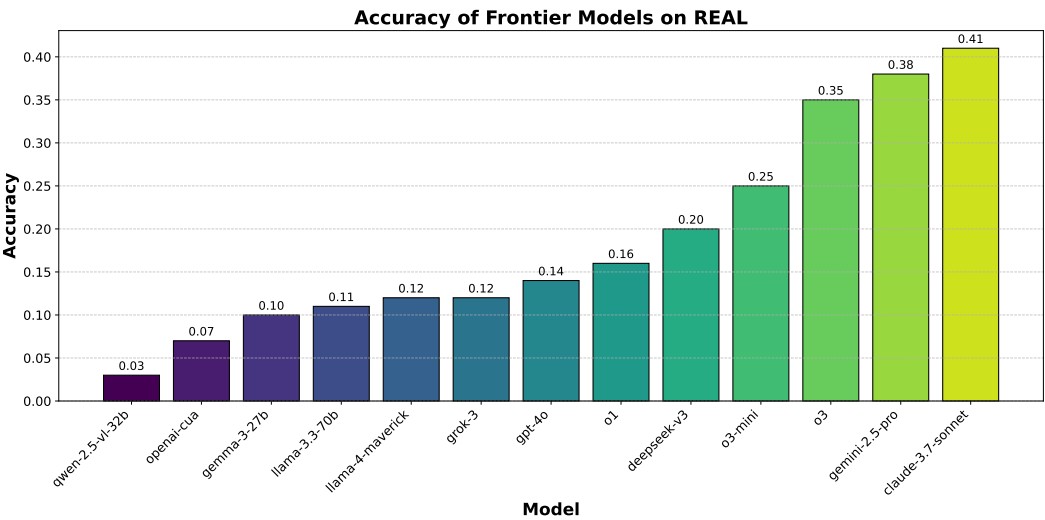

Figure 4: Performance of evaluated models on the REAL benchmark, measured by end-to-end task success rate of our baseline agent across 112 tasks. Claude 3.7 Sonnet-Thinking achieves 41.07%.

We also ran a human baseline with a strict 3-minute limit per task and no prior information on what sites are being used. Participants achieved 96.42% accuracy, indicating the tasks are readily solvable by humans and providing a strong reference point for agent performance.

Overall, our results demonstrate that reliable, autonomous navigation of websites and completion of tasks remains a significant challenge for current frontier models. Similar results are also observed across benchmarks, as studied by [9]. We expect performance to go up with better agent scaffolds beyond our baseline, that integrate search and post-training similar to [40, 46]. REAL is flexible enough to develop harder tasks on the same environments if agents saturate the current test-set.

## 7.1 Qualitative Observations

We analyze interaction traces and outline common failure modes and suggestions below.

**Inadequate Failure Recognition and State Verification.** Agents often fail to assess whether they have successfully completed all parts of the task, lending more weight to their perceived previous actions than the actual updated observation space. For example, within Omnizon, an agent tasked with adding two items to the cart might add the first item but fail to add the second item due to clicking an incorrect button. Despite the cart only containing one item, the agent proceeds through checkout, concluding the interaction under the false assumption that the task was complete. State-verification against the overall goal and error direction thus remain challenging, as also observed in [60, 29].

**Navigation Dead Ends and Lack of Recovery.** Agents often struggle when encountering non-standard navigation flows or unexpected states, and lack the intuition to backtrack effectively. For

example, in the Udriver ride-booking environment, an agent might correctly initiate a booking but then click on an option to schedule the ride for a future time, entering a sub-menu. Once in this sub-environment, agents frequently fail to identify the correct UI element (e.g. a back button, cancel option, or the intended next step) to return to the primary task.

**Guiding the Next Generation of Agents.** REAL ultimately provides highly realistic environments to simulate tasks that are useful and economically valuable in the real-world. Along with this, it can allow for several edge-case configurations that can allow reliability testing on important tasks. Moreover, our sites can be used as RL training environments by setting rewards based on state-diffs or retrieved information, thus providing an important path forward to improve agentic systems.

## 8   Discussion and Future Work

In this work, we introduced REAL, a benchmark and framework designed to evaluate and improve the accuracy and reliability of autonomous web agents in high-fidelity web environments and realistic multi-turn tasks. Our flexible agent harness supports both high-level (Playright) and low-level (CDP) interactions for open and proprietary systems, alongside publicly hosted directly accessible environments that reduce the barrier to entry and facilitates standardized, comparative research.

Beyond its primary role as an evaluation benchmark, REAL is designed to serve as a valuable environment for data generation and agent post-training. Our environments support trajectory collection with rich observation spaces and easy to define rewards, allowing researchers to work on post-training [43, 40, 61], advanced planning [16, 19] and reasoning[49], multi-agent [37], or tree-search methods [25]. Researchers can readily extend the benchmark by defining new tasks with custom goals and evaluation metrics tailored to specific training objectives. Tasks can be chained in future work by composing existing tasks into a multi-stage goal where the end state of one task (e.g., a cart filled, a reservation created, an email drafted) can become the starting condition for the next. Such compositional methods can also allow for long-horizon training [36]. These factors allow REAL to substantially improve over prior work [52, 60, 9].

Our benchmark evaluations on state-of-the-art foundation models highlight substantial room for improvement in agentic performance [29], showcasing REAL as a frontier benchmark for agentic capability research. Although REAL is currently limited to only outcome rewards and a relatively small suite of evaluation tasks, it is easily extensible. As agentic capabilities grow, we will extend REAL with more difficult tasks, long-horizon reasoning [8] or cross-application workflows [14, 5, 50]. Future work will also include a dedicated set of RL post-training workflows to improve agents [27, 12].

REAL delivers an important, rigorous, and accessible framework designed to bridge the gap between current research and practical deployment. Our goal is to drive the development of autonomous agents (Putta et al., 2024), and REAL provides the benchmark and framework necessary to evaluate and train these systems to improve their capability and reliability for important real-world applications.

## Acknowledgements

We would like to thank Milind Maiti, Harshit Sikchi, Julia Kiseleva, Dylan Bowman, Jack Bai, Andrew Gritsevskiy, Matthew Tang, and Lucas Vium for valuable discussions. The websites and configurations are designed under the principles of fair use to serve as tools for research and development. We also note here that Div Garg and Shaun VanWeelden are joint-first authors. Due to issues with setting up OpenReview accounts prior to the conference deadline, we were unable to update the author list. Our public arXiv version reflects the correct order of authors for this work. AD was supported by the EU RRF projects Language Technology Initiative No. 2.3.1.1.i.0/1/22/I/CFLA/002 and Latvian Quantum Initiative No. 2.3.1.1.i.0/1/22/I/CFLA/001.

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

# A Appendix

## A.1 Disclaimer

We will aim to keep improving the benchmark, test suite, and training environment in the near future and have strived to acknowledge the enormous strides made by past work in the area. The authors expressly disclaim any affiliation, association, authorization, endorsement by, or official connection with the real-world companies, brands, or entities represented by the simulated websites. This work constitutes a transformative use under fair use doctrine as it: (1) serves purely academic and non-commercial research purposes; (2) substantially transforms the original material through deterministic behavior and controlled environments; (3) utilizes only the necessary interface elements required for legitimate scientific inquiry; and (4) creates no market substitution for the original websites. Results and evaluations conducted on the Platform are for testing and benchmarking purposes only and should not be construed as equivalent to performing actions on actual websites or applications. While we strive to provide realistic simulations. Any similarities to real-world counterparts are intended solely to replicate core interaction flows in a controlled environment for scientific advancement and do not represent the full functionality, appearance, or underlying technology of the actual websites. All data within these environments is synthetic and does not represent actual information from any commercial platform.

## A.2 Additional Related Work

In addition to benchmarks focusing on everyday tasks, there has also been work focusing on specific use-cases and different dimensions of evaluation. WorkArena [14] and WorkArena++ [3] introduced benchmarks for web agents in the enterprise software setting. AgentBench [33] is broader in that it includes multiple interactive agentic environments (web browsing, code, gaming, etc.), with the goal of providing insights into more general agent capabilities of LLMs. ST-WebAgentBench [28] focused on safety and trustworthiness of web agents, and on assessing web agents' compliance with organizational policies and safety requirements in enterprise settings.

## A.3 Website Tech Stack

REAL website environments are implemented using a modern front-end stack centered on React and Next.js. To ensure consistency across environments, each Next.js project utilizes TypeScript and uses the "app" router configuration. User interface components are derived from the Material UI React library. Critically, all websites are publicly deployed via Vercel, ensuring unrestricted internet accessibility without authentication. This public hosting approach eliminates the setup complexities often associated with prior benchmarks requiring local deployment (e.g., via Docker) [23, 60, 50], thereby lowering the barrier for adoption and facilitating wider research community access. This accessibility setup we provide also reflects the likely operational environment for commercial AI systems designed to interact with public web resources [10, 18, 34, 48].

# B Agent Harness

The REAL Agent Harness provides a standardized interface for evaluating varied agent implementations [53, 10, 40, 43, 24, 51] with minimal required modification. Our goal is to prioritize simplicity and compatibility, enabling researchers to evaluate agents across multiple interaction paradigms while maintaining their existing agent architectures. This approach reduces the technical overhead associated with benchmarking, promoting broader adoption and research across academia and industry.

## B.1 Technical Architecture

The harness offers three integration settings to accommodate different types of agent architectures [1]. As discussed in Section 4, direct Playwright integration grants the user access to a Playwright Browser instance, which enables high-level control of BrowserContext and Page objects for standard web interaction primitives (navigation, element interaction, DOM inspection). For agents requiring lower-level control, the harness provides a WebSocket endpoint for the Chrome DevTools Protocol (CDP),

which allows direct execution of CDP commands across domains like DOM, Runtime, Network, and Input for fine-grained state manipulation. Third, for agents employing black-box systems, our harness supports integration via URL endpoints that expose the browser instance, allowing external controllers to attach and manage the session.

## B.2 Evaluation Flow

A task is initialized when the harness receives a task definition, including a natural language goal ($i$) and a configuration URL. The harness launches and manages a dedicated browser instance, navigating it to the specified /config endpoint (Section 4). Subsequently, control is passed to the agent via its selected integration setting and the agent then enters an iterative loop, receiving observations $o_t$ and executing actions $a_t$ which get translated to corresponding API calls (e.g., page.click(), page.evaluate(), or Input.dispatchKeyEvent via CDP). This interaction cycle continues until the agent attempts to fulfill the task goal $i$, potentially constrained by a maximum step limit. Task completion is signaled by the agent navigating to the designated /submit endpoint or just returning an output/ending the loop (for local client-side evaluation). The harness intercepts this final step, and captures two primary details: the final localStorage state and any agent generated text response (optional), passed via the URL query string. These are then programmatically passed to the task-specific evaluation (outcome reward) functions we describe in Section 5.

## B.3 Submissions and Leaderboard API

Our evaluation framework operates in two modes. A local evaluation returns results directly, allowing quick iterative development and debugging. Researchers can also use the /finish endpoint during these runs to inspect the intermediate localStorage state-diff without concluding the evaluation. Alternatively, navigating to /submit with the correct id can be used for a formal leaderboard submission attempt. This initiates a full evaluation for potential inclusion in public rankings, subject to manual verification. This supports both private research and verified public benchmarking, similar to [60, 56, 14, 47]. We provide a full leaderboard of current frontier language models tested on REAL on our website, and present the accuracy of several of these in Figure 4.

## B.4 Integration of Custom Agents

The REAL harness is designed as an adapter layer to minimize the effort required to integrate custom agents. Researchers can connect their existing systems, including those with proprietary reasoning or planning modules, by implementing the interaction logic against just one of the provided interfaces (Playwright API, CDP command execution, or external control via URL access). This significantly reduces the need for agent-side architectural modifications, lowering the barrier to participation for academic and commercial teams while enabling standardized benchmarking.

## B.5 Evaluating Frontier Agent Frameworks

REAL supports the east integration of external agent frameworks such as Anthropic's Computer Use Agent or OpenAI's Computer-Using Agent. Along with our results based on frontier models using our baseline agent in Figure 4, we also benchmark several frontier agentic scaffolds, both open source (Stagehand and Browseruse) and black-box (Anthropic Computer Use and OpenAI-CUA). Of these, Anthropic's Computer Use Agent recorded the highest score of $42.90\%$, which is very close to the highest score of a frontier model (Claude-3.7-Sonnet-Thinking) with our baseline agent, validating its use as an important starting point to access our REAL benchmark.

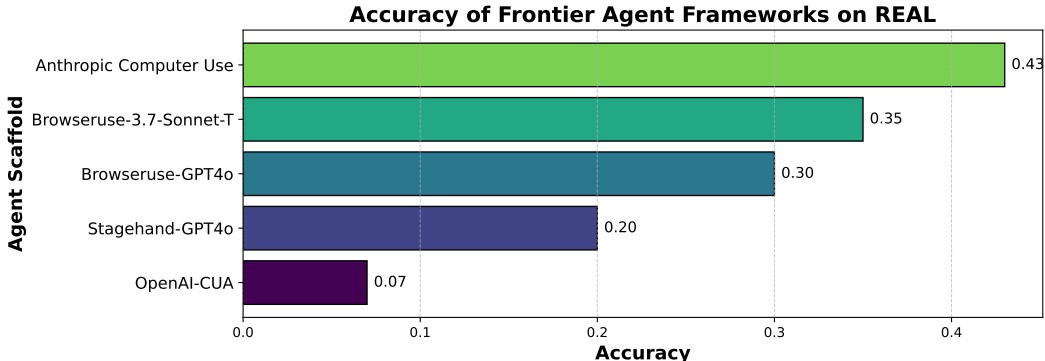

Figure 5: Accuracy of frontier AI Agent frameworks (both open and closed source) on REAL. Anthropic Computer Use Agent achieves state-of-the-art performance at $42.90\%$.

