# OpenReview forum: "REAL: Benchmarking Autonomous Agents on Deterministic Simulations of Real Websites"
_NeurIPS.cc/2025/Datasets_and_Benchmarks_Track — NeurIPS 2025 Datasets and Benchmarks Track poster_

### Official Review · Reviewer_ko8L · 2025-07-02

**Rating:** 5
**Confidence:** 4

**Summary:**

The authors introduce REAL, a benchmark designed to evaluate multi-turn web agents through simulations of real-world websites. The benchmark includes simulations of 11 widely-used websites and 112 practical tasks, reflecting typical daily user interactions. For evaluation, the authors employ a combination of rule-based exact matching and LLM-as-judge assessment. They benchmark 6 leading closed-source models and 6 open-source models on REAL, demonstrating the benchmark's challenging nature and its effectiveness in assessing real-world web agent performance.

**Dataset Code Accessibility:**

Partly

**Ethical Considerations:**

No, there are no or only very minor ethics concerns

**Final Justification:**

Excellent work.
My major concern is that the online services may lack maintenance, harming their long-term availability.
The authors claim that they will make the core code and framework publicly available, thus I raise my confidence accordingly.

**Limitations Weaknesses:**

1. With only 112 evaluation tasks, REAL’s scope is relatively small compared to benchmarks like WebArena (812 intents), potentially restricting comprehensive model assessment.
2. The absence of detailed deployment guides for local website simulations may hinder long-term reproducibility if the hosted version becomes unavailable.
3. Key statistics—such as benchmark runtime, cost, or average interaction turns per task—are missing. Including these details would provide better insight into the practical usability of the benchmark.

**Strengths Contributions:**

1. The benchmark includes 11 simulated websites that are safe, deterministic, and highly realistic, ensuring high-quality evaluations of web agents.
2. The authors provide pre-configured website simulations, enabling easy setup and streamlined benchmarking for researchers.
3. The simulations allow adjustments for factors like network latency and aria labels, facilitating robust evaluation of web agents under varied conditions.
4. The authors include qualitative observations on common failure modes, offering valuable insights for improving web agent performance.

---

> ### Author Rebuttal · Authors · 2025-07-31
>
> Dear Reviewer,
>
> Thank you for your time and valuable feedback! We appreciate the opportunity to address the points raised and further improve our work.
>
> 1. REAL can be easily expanded to contain a much larger range of (difficult) tasks. More tasks can be constructed by hand, by changing the parameters (configurations) or objectives on an environment, or most importantly, by chaining together existing tasks to make them harder and more useful for frontier agents in the future. For our camera-ready version, we commit to releasing 50 additional hard tasks which can be used to extend our evaluation suite easily, as well as a description of how to chain together existing tasks.
>
> 2. Our websites are hosted live and actively maintained by a set of dedicated developers. Along with this, we have now uploaded the core code and framework on which all our websites are built as an AC comment (as per NeurIPS 2025 rebuttal guidelines). This file will become public as soon as the reviews are released, and we commit to open sourcing it with the camera ready version of our paper. We will release this with a local deployment guide that will further support long-term reproducibility on our benchmark.
>
> 3. The duration of successful tasks is 3.3 minutes on average, with interactions taking an average of 4.6 steps with the best model on our benchmark. Unsuccessful tasks require a longer number of steps, but since current agents frequently do not succeed at recognizing failed trajectories and keep continuing until they timeout, it is difficult to effectively measure the average number of steps on these remaining tasks. We have also conducted human baseline experiments to get further insights into the practical usability of our benchmark. Without prior knowledge about the sites on REAL and limited to 3 minutes per task, humans achieve an accuracy of 96.42%. We also note that for future frontier agents, REAL can be extended to harder tasks with more evaluation cases (we commit to adding 50 additional hard tasks for our camera ready version) or by simply chaining together existing tasks.
>
> We sincerely appreciate the reviewer’s feedback and support of our work, and have aimed to address all the comments above.

---

> > ### Comment · Reviewer_ko8L · 2025-08-07
> >
> > Thanks for your rebuttal. As the score is already high, I will keep my score.

---

### Official Review · Reviewer_YqeA · 2025-07-03

**Rating:** 4
**Confidence:** 3

**Summary:**

This paper introduces REAL, a new benchmark and framework for evaluating autonomous web agents on deterministic, high-fidelity replicas of 11 popular websites. The benchmark includes 112 tasks, such as booking a flight or sending an email. The most advanced language models achieve a success rate of 40%+ on the REAL benchmark.

**Dataset Code Accessibility:**

Yes

**Ethical Considerations:**

No, there are no or only very minor ethics concerns

**Final Justification:**

The authors have addressed some of my concerns like human baselines. But concerns like task diversity and potentially saturated benchmarks are not discussed thoroughly.

**Limitations Weaknesses:**

- 11 websites and 112 tasks are still a narrow slice of the real websites and may provide a biased evaluation. It would be helpful to discuss methods for scaling up the benchmark or environment with minimal human effort.
- While the deterministic nature of the simulations is a strength, it is also a limitation. Real sites change layouts, throttle, geo-block, and require CAPTCHAs. Agents trained solely on deterministic clones may overfit. Is it possible to add dynamic content (news feeds), authentication flows, and cross-site tasks to benchmark long-horizon planning? Or is it possible to offer optional noisy modes that randomize element IDs, insert ads, or simulate network lag to study robustness?
- Current scoring gives pass/fail at task completion; partial-credit or step-level diagnostics would be helpful. Another potential improvement is evaluating other important aspects of agent performance, such as efficiency, robustness, and safety. Reporting success-at-k-steps, action efficiency, and error types can help pinpoint failures rather than relying on a single completion flag.
- Human baseline. It would be great to include timed human-in-the-loop runs to calibrate difficulty. Despite the limited robustness and generalization of current state-of-the-art web agents, their achieving 40%+ success on this benchmark raises legitimate concerns about how long the benchmark will remain a meaningful discriminator of real-world agent performance.

**Strengths Contributions:**

- Deterministic simulation of real websites enables reproducible experiments and eliminates the risks associated with testing agents on live websites.
- The 112 tasks are well-designed and reflect real-world use cases.
- The evaluation framework is easy to use.

---

> ### Author Rebuttal · Authors · 2025-07-31
>
> Dear Reviewer,
>
> Thank you for your time and valuable feedback! We appreciate the opportunity to address the points raised and further improve our work.
>
> 1. REAL can be easily expanded to contain a much larger range of (difficult) tasks. More tasks can be constructed by chaining together existing tasks to make them harder, changing the parameters (configurations) or objectives of an environment, or simply by hand. Chaining together existing tasks would require minimal human effort and will also be useful for evaluating frontier agents in the future. Moreover, with our camera-ready version, we will be open sourcing the core framework used for developing all of our environments. This will allow the community to easily build more environments for web agents using a standard website backbone. Along with this, we commit to releasing 50 additional hard tasks (some of which will be chained) which can be used to extend our evaluation suite conveniently.
>
> 2. REAL has already been built with configurability in mind. Our existing environments can be easily configured with several options such as controlled error states, simulating different network latencies, application specific logic parameters such as discounts, etc. via simply appending configuration parameters to the URLs as discussed in Section 6 (dedicated entirely to Configurable Environments). This gives users control over more granular elements throughout our environments. It is also possible to inject dynamic content into environments as has been shown in works such as [1]. We do note that while stochastic environments are important, they also greatly increase computational demands. REAL has therefore been built as a deterministic benchmark at its core, but allows for stochastic extensions via configuration parameters.
>
> 3. We already provide partial credit assignment in a range of existing tasks that are part of our evaluation suite. While REAL distills these subtasks into a binary metric of whether the agent succeeded at the entire task or not, complete metrics with partial rewards are available and accessible via our json files (see for e.g. dashdish-2.json). Importantly, most tasks on REAL are composed of multiple subtasks, which can indeed act as checkpoints to observe how much of the task has been completed along with the binary outcome reward. We also provide default logging for agents that are run on our benchmark, which allows researchers to evaluate granular subtask completion rates and diagnose any issues. We will make this more clear with detailed instructions about accessing partial credit metrics in the camera-ready version of our paper.
>
> 4. We agree completely that human baselines are important to contextualize the performance of agents. As such, we have run human baselines and with a time limit of 3 minutes and no prior knowledge about the sites on REAL, an average accuracy of 96.42% was achieved on our benchmark. This validates that our tasks are logical and reasonably solvable.
>
> [1] WABER: Evaluating Reliability and Efficiency of Web Agents with Existing Benchmarks, Kara et. al.
>
> We sincerely appreciate the reviewer’s feedback and have aimed to address all the comments above. We would be very grateful if the reviewer would consider raising their score further.

---

> > ### Author Response · Authors · 2025-08-05
> >
> > Dear Reviewer,
> >
> > We sincerely appreciate your feedback and have done our best to properly address all your comments. Do let us know if you have any additional questions. We would be very grateful for your consideration in updating our score, thank you!

---

> > ### Comment · Reviewer_YqeA · 2025-08-06
> >
> > I appreciate the clarifications provided, but I still have a few additional points I hope you can further clarify:
> > - While I agree that chaining existing tasks is an effective way to expand tasks, it inherently remains biased towards the domains of current tasks. Also, can you clarify more explicitly the process involved in manually expanding new tasks? What is the procedure to implement automated evaluation and ensure it is correct? Moreover, if one intends to incorporate a completely new website into the benchmark, is it possible to reuse the existing pipeline?
> > - You mentioned that most tasks on REAL consist of multiple subtasks. Can you please specify the proportion of the current tasks that already involve chained subtasks? This information would be helpful in evaluating the diversity and comprehensiveness of the benchmark.
> > - Regarding the human baseline, can you provide more detailed information on how these experiments were conducted, such as participant demographics, instructions given to participants, and the setup of the experimental environment?
> > - It would provide more insights if the revised version could include an analysis of how specific configuration options influence agent performance.

---

> > > ### Author Response · Authors · 2025-08-07
> > >
> > > Dear Reviewer,
> > >
> > > Thank you for providing us the opportunity to provide further details about our benchmark.
> > >
> > > 1. We will be providing two additional ways to expand tasks, the first will consist of 50 additional hard tasks as discussed above and the second will be a path to chain existing tasks. Combined with our preexisting tasks (which are already very diverse and consist of several subtasks as described below), this would provide a large and very diverse range of tasks for evaluations. We are happy to provide more details about how our current tasks were constructed. Our 112 tasks were manually created by a set of annotators who were familiar with the websites. An evaluation was marked as verified only upon independent agreement between all annotators (5 in total) and checking and saving the end state snapshots for a deterministic agreement metric. In a manner consistent with [1], the annotators were asked to create high level intents for websites followed by template based methods of creating our actual evaluation tasks (using annotation guidelines in a manner similar to other popular benchmarks). Our tasks are classified into action based tasks, information retrieval tasks, and tasks that combine goals from both of these settings. Such a method of high-level strategies and low-level template driven tasks makes it very convenient to incorporate new tasks into the benchmark, and even allows for the use of frontier models to support the task generation process if needed in the future [2]. As for incorporating new websites, that is completely possible! We will be open sourcing the core framework used for developing all of our environments as discussed in our previous comment, and this can be used to develop a diverse range of new websites easily with the existing pipeline. Given that our benchmark has an active set of maintainers and developers and the flexibility of our evaluation harness, we will actively aim to incorporate new environments from the open source community.
> > >
> > > 2. Absolutely, we are happy to share more details about the proportion of tasks with multiple subtasks. Currently, 63 of the 112 subtasks have several subtasks which are evaluated, and only if all of these are correct is the complete task verified as correct. Along with this, the individual subtask’s metrics are always accessible. A more granular breakdown of this is: 16 tasks with 2 subtask evaluations, 15 tasks with 3 subtask evaluations, 14 tasks with 4 subtask evaluations, 9 tasks with 5 subtask evaluations, 4 tasks with 6 subtask evaluations, 3 tasks with 7 subtask evaluations, 1 task with 8 subtask evaluations, and 1 with 17 (representative of future longer horizon hard tasks). This represents 56.3% of all our current tasks and we have aimed to create a diverse and comprehensive set of tasks at varying levels of difficulty and length.
> > >
> > > 3. Sure, we recruited 3 participants through a standard annotation service. Each participant was given 3 minutes per task and provided no information about the synthetic websites part of the benchmark beforehand. Our setup consisted of the participants solving the tasks on the benchmark where the tasks (trajectories) would then be evaluated with our evaluation harness. The instructions consisted of the same task goal given to agents along with general instructions on how to access these tasks and the benchmark for the evaluation process in a minimalistic setup.
> > >
> > > 4. We would be happy to discuss in more detail how agents deal with specific configuration options that affect latency, controlled error states, or the availability of items. We will provide this analysis in our camera-ready version using the same evaluation scaffold and configuration techniques that exist currently as part of our benchmark.
> > >
> > > We sincerely appreciate the reviewer’s feedback and the opportunity to clarify our work in more detail! We would be very grateful if the reviewer would consider raising their score.
> > >
> > > [1] WebArena: A Realistic Web Environment for Building Autonomous Agents, Zhou et. al.
> > >
> > > [2] Mind2Web: Towards a Generalist Agent for the Web, Deng et. al.

---

> > > > ### Comment · Reviewer_YqeA · 2025-08-07
> > > >
> > > > Thank you for addressing my previous concerns. I will increase my score to 4. However, since it is not possible to evaluate the 50 additional tasks that the authors promised to include in the future, it is difficult to judge how this can address the task diversity. In addition, I would like the authors to further discuss two additional questions:
> > > > 1. Among the tasks that include subtasks, how many of these subtasks are duplicated?
> > > > 2. Given that current web agents can achieve 40%+ success on this benchmark, how long will the benchmark remain unsaturated and meaningful for real-world agent performance?

---

> > > > > ### Author Response · Authors · 2025-08-09
> > > > >
> > > > > Dear Reviewer,
> > > > >
> > > > > Thank you for increasing your score! We’re happy to answer further questions.
> > > > >
> > > > > 1. Among the tasks that include subtasks, none of the subtasks are duplicated. This ensures that each task is sufficiently diverse, and combined, the benchmark can therefore accurately measure agent capabilities.
> > > > >
> > > > > 2. We expect our benchmark to remain unsaturated for a reasonably long period of time. Our very recent GPT-5 evaluations showed that it was only able to complete 43 of the 112 tasks. Progress on agentic benchmarks has historically taken time and required several new training and scaling techniques. We expect REAL to therefore continue to be challenging for frontier models at least for the foreseeable future, and it remains flexible enough to be updated with more difficult tasks at any point later. Moreover, even as agents improve in the future, REAL will constantly serve as an important measure of their reliability on realistic web tasks.

---

### Official Review · Reviewer_ugQU · 2025-07-03

**Rating:** 5
**Confidence:** 4

**Summary:**

This paper introduces REAL, a new benchmark and framework for evaluating autonomous web agents. Its core contributions include:
- A collection of $11$ high-fidelity, deterministic, and publicly hosted simulations of popular real-world websites across various domains like e-commerce, travel, and communication.
- A novel evaluation framework that combines programmatic checks of website state changes (via localStorage) for action-based tasks with rubric-guided LLM judgments for information retrieval tasks.
- A flexible and accessible agent harness that supports various agent architectures (including open-source and proprietary systems) through multiple interaction methods like Playwright and the Chrome DevTools Protocol (CDP), lowering the barrier to entry for researchers.
- A benchmark suite of 112 practical, multi-turn tasks designed to mirror everyday user interactions, covering both information retrieval and state-changing actions.

**Dataset Code Accessibility:**

Yes

**Dataset Code Comments:**

The authors have done an excellent job of making their work accessible and usable. They provide well-documented open-source code and assets in a GitHub repository. The inclusion of clear examples and publicly hosted websites greatly facilitates adoption and reproducibility.

**Ethical Considerations:**

No, there are no or only very minor ethics concerns

**Limitations Weaknesses:**

The primary weakness of the benchmark in its current form relates to task difficulty.
- **Lack of Human Baseline**: This paper does not seem to include a formal human evaluation or performance baseline for its tasks. While the tasks are designed to mirror real user goals, a human baseline would be invaluable for calibrating the benchmark's difficulty. It would help to contextualize agent performance (e.g., is $41\\%$ success good or bad compared to a human?) and validate that the tasks are logical and reasonably solvable.
- **Moderate Task Difficulty**: The evaluation results indicate that the current suite of $112$ tasks may not be sufficiently challenging to push the limits of the most advanced models. State-of-the-art models can achieve success rates of over $40\\%$ using a standard baseline agent, without agent-specific optimizations. While this demonstrates the benchmark is not yet "solved," it suggests that top models may saturate performance on the current tasks relatively quickly, which may limit the benchmark's long-term utility for differentiating frontier agents unless more difficult tasks are added.

**Strengths Contributions:**

The REAL benchmark presents several significant strengths that advance the state of web agent evaluation.
- **Diverse and High-Fidelity Environments**: The benchmark includes 11 distinct website replicas, offering a more diverse set of domains and functionalities compared to many prior benchmarks. These environments are built with a modern tech stack to ensure high visual and functional fidelity, providing a realistic testing ground.
- **Excellent Reproducibility and Accessibility**: A major strength is the focus on determinism. By fixing data, timestamps, and UX elements, REAL ensures that evaluations are robust and reproducible. The decision reduces the overhead of local setup and costs, making the benchmark highly accessible to the broader research community.
- **Designed for Automated Evaluation**: The framework is thoughtfully designed for ease of use and automation. The inclusion of dedicated URL endpoints, such as `/config` for setup, `/submit` for evaluation, and `/clear` for resetting the state, simplifies the process of running programmatic assessments at scale.

---

> ### Author Rebuttal · Authors · 2025-07-31
>
> Dear Reviewer,
>
> Thank you for your time and valuable feedback! We appreciate the opportunity to address the points raised and further improve our work.
>
> 1. We agree completely that human baselines are important to contextualize the performance of agents. As such, we have run human baselines and with a time limit of 3 minutes and no prior knowledge about the sites on REAL, an average accuracy of 96.42% was achieved on our benchmark. This validates that our tasks are logical and reasonably solvable, and we will discuss this result further in our camera-ready version.
>
> 2. REAL can be expanded to contain a much larger range of (difficult) tasks. More tasks can be constructed by hand, by changing the parameters (configurations) or objectives on an environment, and most importantly, by chaining together existing tasks to make them harder and more useful for frontier agents in the future. For our camera-ready version, we commit to releasing 50 additional hard tasks which can be used to extend our evaluation suite easily, as well as a description of how to chain together existing tasks.
>
> We sincerely appreciate the reviewer’s feedback and support of our work, and have aimed to address all the comments above.

---

> > ### Comment · Reviewer_ugQU · 2025-08-03
> >
> > Great - thank you for your responses. As my score is already high, I leave it as is.

---

### Official Review · Reviewer_7TG4 · 2025-07-03

**Rating:** 5
**Confidence:** 4

**Summary:**

REAL is a toolkit for building and evaluating browser-based AI agents in realistic, reproducible environments. It powers realevals.xyz — a public leaderboard and evaluation platform for agents navigating complex web apps, including replicas of Amazon, DoorDash, Airbnb, and more. It supports 1) train and benchmark agents with robust, standardized tasks, 2) use plug-and-play LLMs or your own agent logic, 3) evaluate capabilities via deterministic simulations, ensuring scientific reproducibility.

**Dataset Code Accessibility:**

Partly

**Dataset Code Comments:**

The environment websites are not publicly available.

**Ethical Considerations:**

No, there are no or only very minor ethics concerns

**Final Justification:**

After releasing the website and all the self-hosting code, the work become more valuable for researchers.

**Limitations Weaknesses:**

1. The biggest weakness is the the key contribution, the websites themselves are publicly hosted by the authors, instead of distributed in a fashion where researchers can self host. If the public website goes down, all the research on top of this benchmark will no longer be available for comparison and improvement.
2. Similar to above, the non public nature also means that people cannot easily modify the state, reset the website or self host for larger training.
3. Even though the models tested are extensive, it'd be more complete and robust to test another agent harnesses other than the provided one. As a benchmark and environment, it should ideally be agnostic to the actual agent harness used, as that is part of research on agents moving forward.

**Strengths Contributions:**

1. The benchmark is well constructed with "fake" websites that mimic realistic websites populated with realistic "fake" data. This sandboxing approach is proven to be effective for agent evaluation and training since hitting public live websites with these requests are not only risky but also barebones malicious. The websites themselves poses a huge engineering efforts, giving the benchmark a lot of credits
2. The tasks themselves is diverse and utilizes several websites. Compared with existing work such as WebArena, VisualWebArena, this work covers a different set of simulated websites that closes the gap in the field of web agent research by improving the coverage of the sandboxed internet concepts. Also, these websites are built with UI elements and design language, UX similar to popular live websites such as Amazon, Gmail, DoorDash, etc. and further closing the sim2real gap.
3. Configurable website environments allow for better control of how the user want to present these website UI/UX logic, etc.
4. Extensive experiments on both closed models and open models.

---

> ### Author Rebuttal · Authors · 2025-07-31
>
> Dear Reviewer,
>
> Thank you for your time and valuable feedback! We appreciate the opportunity to address the points raised and further improve our work.
>
> 1. Our websites are hosted live and are actively maintained by a team of dedicated developers. Along with this, we have now uploaded the core code and framework on which our websites are built as an AC comment (as per NeurIPS 2025 rebuttal guidelines). This file will become public as soon as the reviews are released, and we commit to open sourcing it with the camera ready version of our paper.
>
> 2. Our existing environments can be easily configured with several options such as controlled error states, simulating different latencies, application-specific logic parameters, regional contexts, etc. as described in Section 6. Moreover, the state of any website can be reset locally by navigating to ‘\clear’ as discussed in Sections 3 and 4. With these settings available, our benchmark has been built with configurability in mind. Moreover, our core code website backbone code (described above) is easily modifiable and configurable for more significant changes or new self-hosted environments.
>
> 3. Along with our default agent harness with which we evaluate a range of open and closed source models, we provide several other agent harness implementations and evaluation results. Appendix B.5 contains a description of these harnesses, which include Anthropic and OpenAI Computer Use harnesses along with Browseruse and Stagehand. Our environments are therefore agnostic to harnesses and we’ve provided experimental results with a range of setups.
>
> We sincerely appreciate the reviewer’s feedback and have aimed to address all the comments above. We would be grateful if the reviewer would consider raising their score further.

---

> > ### Author Response · Authors · 2025-08-05
> >
> > Dear Reviewer,
> >
> > We sincerely appreciate your feedback and have done our best to properly address all your comments. Do let us know if you have any additional questions. We would be very grateful for your consideration in updating our score, thank you!

---

> > > ### Comment · Reviewer_7TG4 · 2025-08-06
> > >
> > > Thank you for your responses. If the websites themselves become open source and fully self-hostable and reproducible, yes, it deserves a higher score. I have raised my score.

---

### Decision · Program_Chairs · 2025-09-18

**Decision:**

Accept (poster)

**Comment:**

First, the paper is pretty clear in what it contributes: REAL is a benchmark/framework for evaluating browser-based agents in deterministic but realistic environments. It includes 11 high-fidelity replica websites (Amazon, DoorDash, Airbnb, Gmail-like apps, etc.) and 112 tasks, covering both retrieval and action-based interactions. It also introduces an evaluation pipeline (mixing programmatic checks with LLM judgments), and provides a harness supporting both closed and open source models. The experiments are thorough: multiple strong baselines are run, both commercial and open-source, and they also share qualitative observations. (best models achieve ~40%+ success).

On strengths, I think the reviewers are right to emphasize the engineering lift here. Building these websites to look/feel like Amazon/DoorDash while keeping them safe and reproducible is non-trivial and genuinely valuable. The determinism is a big plus — reproducibility is a constant problem in this space, so freezing timestamps, data, UX states, etc. makes this benchmark really useful to the community. The accessibility angle (publicly hosted, easy reset/config, plug-and-play harnesses) means more researchers can actually use it without heavy infra, which fits DB track’s spirit of enabling systematic, large-scale evaluation. And compared with prior work (WebArena, VisualWebArena, etc.), this closes an important gap: new domains, more realistic UI/UX, and tighter sim2real fidelity.

On weaknesses, I’ll echo the reviewers’ joint concerns. The biggest is sustainability of the hosted approach — if the hosted servers go down, all reproducibility disappears. Authors do promise open-sourcing the backbone code (per rebuttal), which helps, but I agree that long-term self-hosting / deployment instructions should be more explicit. Another concern is scope: 11 websites / 112 tasks is a good start, but small compared to other benchmarks, so scalability is a question. Some reviewers also asked about robustness — right now everything is deterministic, but real websites are noisy (ads, throttling, authentication, CAPTCHAs). Without any “noisy” modes, there’s a risk of overfitting. Finally, evaluation granularity could be richer than just pass/fail (step-level diagnostics, partial credit, efficiency metrics, safety dimensions, etc.). And yes, a human baseline would have been important (though the rebuttal says they added one, which helps a lot).